# Todyformer: Towards Holistic Dynamic Graph Transformers with Structure-Aware Tokenization

## Abstract

Temporal Graph Neural Networks have garnered substantial attention for their capacity to model evolving structural and temporal patterns while exhibiting impressive performance. However, it is known that these architectures are encumbered by issues that constrain their performance, such as over-squashing and over-smoothing. Meanwhile, Transformers have demonstrated exceptional computational capacity to effectively address challenges related to long-range dependencies. Consequently, we introduce Todyformer—a novel Transformer-based neural network tailored for dynamic graphs. It unifies the local encoding capacity of Message-Passing Neural Networks (MPNNs) with the global encoding of Transformers through i) a novel patchifying paradigm for dynamic graphs to improve over-squashing, ii) a structure-aware parametric tokenization strategy leveraging MPNNs, iii) a Transformer with temporal positional-encoding to capture long-range dependencies, and iv) an encoding architecture that alternates between local and global contextualization, mitigating over-smoothing in MPNNs. Experimental evaluations on public benchmark datasets demonstrate that Todyformer consistently outperforms the state-of-the-art methods for downstream tasks. Furthermore, we illustrate the underlying aspects of the proposed model in effectively capturing extensive temporal dependencies in dynamic graphs.

## 1 Introduction

Dynamic graphs, driven by the surge of large-scale structured data on the internet, have become pivotal in graph representation learning. Dynamic graphs are simply static graphs where edges have time attributes Kazemi et al. (2020). Representation learning approaches for dynamic graphs aim to learn how to effectively encode recurring structural and temporal patterns for node-level downstream tasks. For instance, Future Link Prediction (FLP) uses past interactions to predict future links, while Dynamic Node Classification (DNC) focuses on predicting labels of upcoming nodes based on impending interactions. While models based on Message-Passing Neural Networks (MPNN) Gilmer et al. (2017) have demonstrated impressive performance on encoding dynamic graphs Rossi et al. (2020); Wang et al. (2021); Jin et al. (2022); Luo & Li (2022), many approaches have notable limitations. Primarily, these methods often rely heavily on chronological training or use complex memory modules for predictions Kumar et al. (2019); Xu et al. (2020); Rossi et al. (2020); Wang et al. (2021), leading to significant computational overhead, especially for dynamic graphs with many edges. Additionally, the use of inefficient message-passing procedures can be problematic, and some methods depend on computationally expensive random-walk-based algorithms Wang et al. (2021); Jin et al. (2022). These methods often require heuristic feature engineering, which is specifically tailored for edge-level tasks.

Moreover, there is a growing consensus within the community that the message-passing paradigm is inherently constrained by the hard inductive biases imposed by the graph structure Kreuzer et al. (2021). A central concern with conventional MPNNs revolves around the over-smoothing problem stemmed from the exponential growth of the model's computation graph Dwivedi & Bresson (2020). This issue becomes pronounced when the model attempts to capture the higher-order long-range aspects of the graph structure. Over-smoothing hurts model expressiveness in MPNNs where the network depth grows in an attempt to increase expressiveness. However, the node embeddings

tend to converge towards a constant uninformative representation. This serves as a reminder of the lack of flexibility observed in early recurrent neural networks used in Natural Language Processing (NLP), especially when encoding lengthy sentences or attempting to capture long-range dependencies within sequences Hochreiter & Schmidhuber (1997). However, Transformers have mitigated these limitations in various data modalities Vaswani et al. (2017); Devlin et al. (2018); Liu et al. (2021); Dosovitskiy et al. (2020); Dwivedi & Bresson (2020). Over-squashing is another problem that message-passing networks suffer from since the amount of local information aggregated repeatedly increases proportionally with the number of edges and nodes Hamilton (2020); Topping et al. (2021).

To address the aforementioned learning challenges on dynamic graphs, we propose Todyformer[1]—a novel Graph Transformer model on dynamic graphs that unifies the local and global message-passing paradigms by introducing patchifying, tokenization, and encoding modules that collectively aim to improve model expressiveness through alleviating over-squashing and over-smoothing in a systematic manner. To mitigate the neighborhood explosion (i.e, over-squashing), we employ temporal-order-preserving patch generation, a mechanism that divides large dynamic graphs into smaller dynamic subgraphs. This approach breaks the larger context into smaller subgraphs suitable for local message-passing, instead of relying on the model to directly analyze the granular and abundant features of large dynamic graphs.

Moreover, we adopt a hybrid approach to successfully encode the long-term contextual information, where we use MPNNs for tasks they excel in, encoding local information, while transformers handle distant contextual dependencies. In other words, our proposed architecture adopts the concept of learnable structure-aware tokenization, reminiscent of the Vision Transformer (ViT) paradigm Dosovitskiy et al. (2020), to mitigate computational overhead. Considering the various contributions of this architecture, Todyformer dynamically alternates between encoding local and global contexts, particularly when capturing information for anchor nodes. This balances between the local and global computational workload and augments the model expressiveness through the successive stacking of the MPNN and Transformer modules.

## 2 RELATED WORK

**Representation learning for dynamic graphs:** Recently, the application of machine learning to Continuous-Time Dynamic Graphs (CTDG) has drawn the attention of the graph community Kazemi et al. (2020). RNN-based methods such as JODIE Divakaran & Mohan (2020), Know-E Trivedi et al. (2017), and DyRep Trivedi et al. (2019) typically update node embeddings sequentially as new edges arrive. TGAT Xu et al. (2020), akin to GraphSAGE Hamilton et al. (2017) and GAT Veličković et al. (2018), uses attention-based message-passing to aggregate messages from historical neighbors of an anchor node. TGN Rossi et al. (2020) augments the message-passing with an RNN-based memory module that stores the history of all nodes with a memory overhead. CAW Wang et al. (2021) and NeurTWs Jin et al. (2022) abandon the common message-passing paradigm by extracting temporal features from temporally-sampled causal walks. CAW operates directly within link streams and mandates the retention of the most recent links, eliminating the need for extensive memory storage. Moreover, Souza et al. (2022) investigates the theoretical underpinnings regarding the representational power of dynamic encoders based on message-passing and temporal random walks. DyG2Vec Alomrani et al. (2022) proposes an efficient attention-based encoder-decoder MPNN that leverages temporal edge encoding and window-based subgraph sampling to regularize the representation learning for task-agnostic node embeddings. GraphMixer Cong et al. (2023) simplifies the design of dynamic GNNs by employing fixed-time encoding functions and leveraging the MLP-Mixer architecture Tolstikhin et al. (2021).

**Graph Transformers:** Transformers have been demonstrating remarkable efficacy across diverse data modalities Vaswani et al. (2017); Dosovitskiy et al. (2020). The graph community has recently started to embrace them in various ways Dwivedi & Bresson (2020). Graph-BERT Zhang et al. (2020) avoids message-passing by mixing up global and relative scales of positional encoding. Kreuzer et al. (2021) proposes a refined inductive bias for Graph Transformers by introducing a soft and learnable positional encoding (PE) rooted in the graph Laplacian domain, signifying a substantive stride in encoding low-level graph structural intricacies. Ying et al. (2021) is provably

---

[1] We are going to open-source the code upon acceptance.

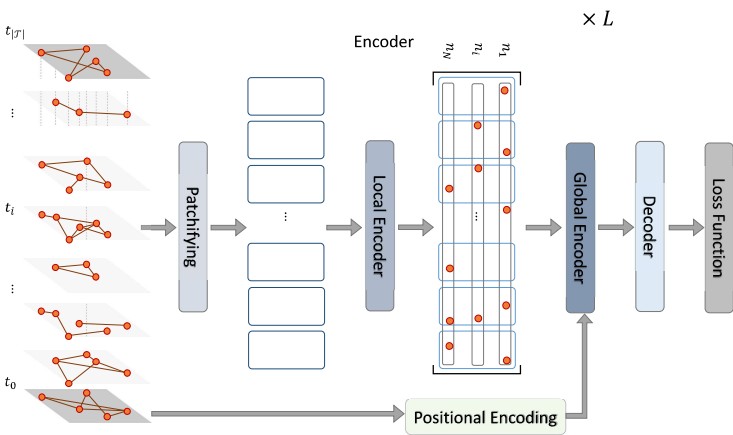

Figure 1: Illustration of Todyformer encoding-decoding architecture.

more powerful than 1-WL; it abandons Laplacian PE in favor of spatial and node centrality PEs. Subsequently, SAT Chen et al. (2022) argues that Transformers with PE do not necessarily capture structural properties. Therefore, the paper proposes applying GNNs to obtain initial node representations. Graph GPS Rampášek et al. (2022) provides a recipe to build scalable Graph Transformers, leveraging structural and positional encoding where MPNNs and Transformers are jointly utilized to address over-smoothing, similar to SAT. TokenGT Kim et al. (2022) demonstrates that standard Transformers, without graph-specific modifications, can yield promising results in graph learning. It treats nodes and edges as independent tokens and augments them with token-wise embeddings to capture the graph structure. He et al. (2023) adapts MLP-Mixer Tolstikhin et al. (2021) architectures to graphs, partitioning the input graph into patches, applying GNNs to each patch, and fusing their information while considering both node and patch PEs. While the literature adapts Transformers to static graphs, a lack of attention is eminent on dynamic graphs. In this work, we strive to shed light on such adaptation in a principled manner and reveal how dynamic graphs can naturally benefit from a unified local and global encoding paradigm.

## 3 Todyformer: Tokenized Dynamic Graph Transformer

We begin this section by presenting the problem formulation of this work. Next, we provide the methodological details of the Todyformer architecture along with its different modules.

**Problem Formulation:** A CTDG $\mathcal{G} = (\mathcal{V}, \mathcal{E}, \mathcal{X}^E, \mathcal{X}^v)$ with $N = |\mathcal{V}|$ nodes and $E = |\mathcal{E}|$ edges can be represented as a sequence of interactions $\mathcal{E} = \{e_1, e_2, \ldots, e_E\}$, where $\mathcal{X}^v \in \mathbb{R}^{N \times D^V}$ and $\mathcal{X}^E \in \mathbb{R}^{E \times D^E}$ are the node and edge features, respectively. $D^V$ and $D^E$ are the dimensions of the node and edge feature space, respectively. An edge $e_i = (u_i, v_i, t_i, m_i)$ links two nodes $u_i, v_i \in \mathcal{V}$ at a continuous timestamp $t_i \in \mathbb{R}$, where $m_i \in \mathcal{X}^E$ is an edge feature vector. Without loss of generality, we assume that the edges are undirected and ordered by time (i.e., $t_i \leq t_{i+1}$). A temporal sub-graph $\mathcal{G}_{ij}$ is defined as a set consisting of all the edges in the interval $[t_i, t_j]$, such that $\mathcal{E}_{ij} = \{e_k \mid t_i \leq t_k < t_j\}$. Any two nodes can interact multiple times throughout the time horizon; therefore, $\mathcal{G}$ is a multi-graph. Following DyG2Vec Alomrani et al. (2022), we adopt a window-based encoding paradigm for dynamic graphs to control representation learning and balance the trade-off between efficiency and accuracy according to the characteristics of the input data domain. The parameter $W$ controls the size of the window for the input graph $\mathcal{G}_{ij}$, where $j = i + W$. For notation brevity, we assume the window mechanism is implicit from the context. Hence, we use $\mathcal{G}$ as the input graph unless explicit clarification is needed.

Based on the downstream task, the objective is to learn the weight parameters $\theta$ and $\gamma$ of a dynamic graph encoder $f_\theta$ and decoder $g_\gamma$ respectively. $f_\theta$ projects the input graph $\mathcal{G}$ to the node embeddings $\boldsymbol{H} \in \mathbb{R}^{N \times D^H}$, capturing temporal and structural dynamics for the nodes. Meanwhile, a decoder $g_\gamma$ outputs the predictions given the node embeddings for the downstream task, enabling accurate

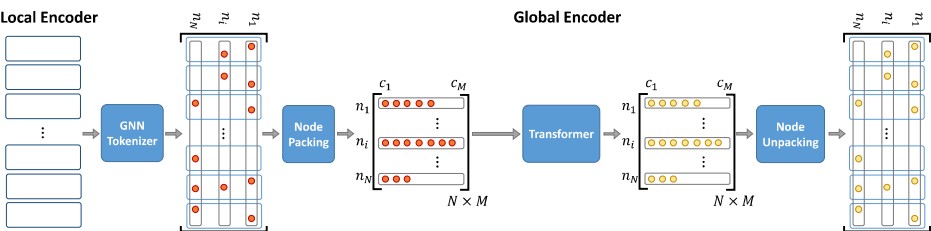

Figure 2: Schematic depiction of the computation flow in the local and global encoding modules, particularly highlighting node packing and unpacking modules in Todyformer.

future predictions based on past interactions. More specifically:

$$\boldsymbol{H} = f_\theta(\mathcal{G})\,, \qquad \boldsymbol{Z} = g_\gamma(\boldsymbol{H})\,, \qquad (1)$$

Here, $\boldsymbol{Z}$ represents predictions for the ground-truth labels. In this work, we focus on common downstream tasks defined similarly to Alomrani et al. (2022) for training and evaluation: i) Future Link Prediction (FLP) and ii) Dynamic Node Classification (DNC).

## 3.1 ENCODER ARCHITECTURE

Todyformer consists of $L$ blocks of encoding $f_\theta = \{(u^l, v^l)\}_{l=0}^L$ where $u = \{u^l\}_{l=0}^L$ and $v = \{v^l\}_{l=0}^L$ are the sets of local and global encoding modules, respectively. As illustrated in Figure 1, the encoding network of Todyformer benefits from an alternating architecture that alternates between local and global message-passing. The local encoding is structural and temporal, according to the learnable tokenizer, and the global encoding in this work is defined to be temporal. We leave the structural and temporal global encoding for future work. In the following, we define each encoding module in more detail.

## 3.2 PATCH GENERATION

Inspired by Dosovitskiy et al. (2020), Todyformer begins by partitioning a graph into $M$ subgraphs, each containing an equal number of edges. This partitioning is performed based on the timestamp associated with each edge. Specifically, the patchifier $p$ evenly segments the input graph $\mathcal{G}$ with $\mathcal{E} = \{e_1, e_2, \ldots, e_E\}$ edges into $M$ non-overlapping subgraphs of equal size, referred to as patches. More concretely:

$$\mathcal{P} = p(\mathcal{G}; M) \qquad (2)$$

where $\mathcal{P} = \{\mathcal{G}_m \,|\, m \in \{1, 2, ..., \frac{E}{M}\}\}$ and the $m$-th graph, denoted as $\mathcal{G}_m$, consists of edges with indices in the range $\{(m-1)\frac{E}{M} + 1, \cdots, m\frac{E}{M}\}$. Partitioning the input graph into $M$ disjoint subgraphs helps message-passing to be completely separated within each patch. Additionally, $M$ manages the trade-off between alleviating over-squashing and maintaining the tokenizer's expressiveness. Through ablation studies, we empirically reveal how different datasets react to various $M$ values.

## 3.3 LOCAL ENCODING: STRUCTURE-AWARE TOKENIZATION

Local encoding $u^l = (s^l, c)$ contains two modules: the tokenization $s^l$ and the packing $c$ modules. The former handles local tokenization, and the latter packs tokens into a sequential data structure that will be consumed by the global encoder.

**Structure-Aware Tokenization:** Following the recent trend in Graph Transformers, where tokenization is structure-aware, local encoding in Todyformer utilizes a dynamic GNN to map the input node embeddings to the latent embeddings that a Transformer will process later on. It should be noted that the tokenizer has learnable parameters to encode both structural and temporal patterns in the patches. Without loss of generality, we use DyG2Vec Alomrani et al. (2022) as a powerful

attentive message-passing model to locally encode input features into semantically meaningful node tokens.

$$\mathcal{H}^l = s^l(\bar{\mathcal{H}}^{l-1}) \tag{3}$$

where $\mathcal{H}^l = \{H_i^l\}_{i=0}^{M-1}$ is the set of node embeddings $H_i$ for $M$ different patches, and $\bar{\mathcal{H}}^{l-1}$ is the set of node embeddings computed by the previous block. As illustrated in Figure 1, the output of one block from the global encoder is transferred as the input into the local encoder of the subsequent block. It should be noted that $\bar{\mathcal{H}}^0 = \mathcal{X}$ for the first layer, where $\mathcal{X} = \{X_i\}_{i=0}^{M-1}$ is the set of node features for all patches.

**Packing:** Once the node features are locally encoded into node tokens, the next step is to pack the set of node embeddings $\mathcal{H}^l$ into the standard format required by Transformers. Since a node may appear in multiple patches, to collect all the node embeddings for a particular node across the patches, a node-packing module $c$ is utilized. This module collects the embeddings of all nodes across the patches and arranges them in a sequential data format as follows:

$$H^l = c(\mathcal{H}^l, \mathcal{P}) \tag{4}$$

where $H^l \in \mathbb{R}^{N \times M \times D^H}$ such that $N$ is the number of nodes in the input graph $\mathcal{G}$, $M$ is the total number of patches, and $D^H$ is the dimension of the embedding space. The module $c$ uses $\mathcal{P}$ to figure out which patches a node belongs to. Consequently, the output of the local encoding module is structured in a tensor that can be easily consumed by a Transformer. The computation flow in the local encoder is shown in Figure 2. Since nodes may have interactions for a variable number of times in the input graph, it is necessary to pad the short sequences with the [MASK] tokens at the end. Then, the mini-batch of sequences can be easily packed into a dense tensor and fed as input to Transformers.

### 3.4 GLOBAL ENCODING

The packed node tokens will be fed into the global encoding module to perform long-range message-passing beyond the local context of the input patches. Therefore, Todyformer not only maximizes the parametric capacity of MPNNs to encode local context but also leverages the long-range capacities of Transformers to improve the model expressiveness. The global encoder $v^l = (o^l, r^l, n^l)$ consists of the positional encoder $o^l$, Transformer $r^l$, and unpacking module $n^l$ according to the details provided in the following.

**Positional Encoding:** Transformers are aware of the ordering in the input sequences through positional encoding. Various systematic approaches have been investigated in the literature for the sake of improved expressiveness (Dwivedi & Bresson, 2020; Kreuzer et al., 2021). Once the structural and temporal features are locally mapped into node tokens, and the sequential input $H^l$ is packed at layer $l$, positional encoding is needed to inform the Transformer of the temporal ordering of the node tokens on a global scale. The positional encoding in Todyformer is defined based on the notion of the position and the encoding function. The position can be explicitly defined as the global edge index of a node upon an interaction at a timestamp or implicitly defined as the patch or occurrence indices. The encoding function can be a linear or sinusoidal mapping. The PE is fused into the packed node embeddings through the addition modulation, as follows:

$$H^l = H^l + P, \qquad P = o(\mathcal{P}) \in \mathbb{R}^{N \times M \times D^H} \tag{5}$$

**Transformer:** The global encoding updates node embeddings through a dot-product Multi-head Self-Attention (MSA) Transformer architecture as follows:

$$\bar{H}^l = r^l(H^l), \qquad r^l = \texttt{Transformer}(Q, K, V) = \text{softmax}\Big(\frac{QK^T}{\sqrt{d_k}}\Big)V \tag{6}$$

where $Q = H^l W_q \in \mathbb{R}^{N \times D^k}$, $K = H^l W_k \in \mathbb{R}^{N \times D^k}$, and $V = H^l W_v \in \mathbb{R}^{N \times D^v}$ are the query, key, and value, respectively, and $W_q, W_k \in \mathbb{R}^{D^H \times D^k}$ and $W_v \in \mathbb{R}^{D^H \times D^v}$ are learnable matrices.

Table 1: Future Link Prediction Performance in AP (Mean ± Std) on the test set.

| Setting | Model | MOOC | LastFM | Enron | UCI | SocialEvol. |
|---|---|---|---|---|---|---|
| Transductive | JODIE | $0.797 \pm 0.01$ | $0.691 \pm 0.010$ | $0.785 \pm 0.020$ | $0.869 \pm 0.010$ | $0.847 \pm 0.014$ |
| | DyRep | $0.840 \pm 0.004$ | $0.683 \pm 0.033$ | $0.795 \pm 0.042$ | $0.524 \pm 0.076$ | $0.885 \pm 0.004$ |
| | TGAT | $0.793 \pm 0.006$ | $0.633 \pm 0.002$ | $0.637 \pm 0.002$ | $0.835 \pm 0.003$ | $0.631 \pm 0.001$ |
| | TGN | $0.911 \pm 0.010$ | $0.743 \pm 0.030$ | $0.866 \pm 0.006$ | $0.843 \pm 0.090$ | $0.966 \pm 0.001$ |
| | CaW | $0.940 \pm 0.014$ | $0.903 \pm 1e\text{-}4$ | $0.970 \pm 0.001$ | $0.939 \pm 0.008$ | $0.947 \pm 1e\text{-}4$ |
| | NAT | $0.874 \pm 0.004$ | $0.859 \pm 1e\text{-}4$ | $0.924 \pm 0.001$ | $0.944 \pm 0.002$ | $0.944 \pm 0.010$ |
| | GraphMixer | $0.835 \pm 0.001$ | $0.862 \pm 0.003$ | $0.824 \pm 0.001$ | $0.932 \pm 0.006$ | $0.935 \pm 3e\text{-}4$ |
| | Dygformer | $0.892 \pm 0.005$ | $0.901 \pm 0.003$ | $0.926 \pm 0.001$ | $0.959 \pm 0.001$ | $0.952 \pm 2e\text{-}4$ |
| | DyG2Vec | $\underline{0.980 \pm 0.002}$ | $\underline{0.960 \pm 1e\text{-}4}$ | $\underline{0.991 \pm 0.001}$ | $\underline{0.988 \pm 0.007}$ | $\underline{0.987 \pm 2e\text{-}4}$ |
| | **Todyformer** | $\mathbf{0.992 \pm 7e\text{-}4}$ | $\mathbf{0.976 \pm 3e\text{-}4}$ | $\mathbf{0.995 \pm 6e\text{-}4}$ | $\mathbf{0.994 \pm 4e\text{-}4}$ | $\mathbf{0.992 \pm 1e\text{-}4}$ |
| Inductive | JODIE | $0.707 \pm 0.029$ | $0.865 \pm 0.03$ | $0.747 \pm 0.041$ | $0.753 \pm 0.011$ | $0.791 \pm 0.031$ |
| | DyRep | $0.723 \pm 0.009$ | $0.869 \pm 0.015$ | $0.666 \pm 0.059$ | $0.437 \pm 0.021$ | $0.904 \pm 3e\text{-}4$ |
| | TGAT | $0.805 \pm 0.006$ | $0.644 \pm 0.002$ | $0.693 \pm 0.004$ | $0.820 \pm 0.005$ | $0.632 \pm 0.005$ |
| | TGN | $0.855 \pm 0.014$ | $0.789 \pm 0.050$ | $0.746 \pm 0.013$ | $0.791 \pm 0.057$ | $0.904 \pm 0.023$ |
| | CaW | $0.933 \pm 0.014$ | $0.890 \pm 0.001$ | $0.962 \pm 0.001$ | $0.931 \pm 0.002$ | $0.950 \pm 1e\text{-}4$ |
| | NAT | $0.832 \pm 1e\text{-}4$ | $0.878 \pm 0.003$ | $0.949 \pm 0.010$ | $0.926 \pm 0.010$ | $0.952 \pm 0.006$ |
| | GraphMixer | $0.814 \pm 0.002$ | $0.821 \pm 0.004$ | $0.758 \pm 0.004$ | $0.911 \pm 0.004$ | $0.918 \pm 6e\text{-}4$ |
| | Dygformer | $0.869 \pm 0.004$ | $0.942 \pm 9e\text{-}4$ | $0.897 \pm 0.003$ | $0.945 \pm 0.001$ | $0.931 \pm 4e\text{-}4$ |
| | DyG2Vec | $\underline{0.938 \pm 0.010}$ | $\underline{0.979 \pm 0.006}$ | $\underline{0.987 \pm 0.004}$ | $\underline{0.976 \pm 0.002}$ | $\underline{0.978 \pm 0.010}$ |
| | **Todyformer** | $\mathbf{0.948 \pm 0.009}$ | $\mathbf{0.981 \pm 0.005}$ | $\mathbf{0.989 \pm 8e\text{-}4}$ | $\mathbf{0.983 \pm 0.002}$ | $\mathbf{0.9821 \pm 0.005}$ |

We apply an attention mask to enforce directed connectivity between node tokens through time, where a node token from the past is connected to all others in the future. The Transformer module is expected to learn temporal inductive biases from the context on how to deploy attention on recent interactions versus early ones.

**Unpacking:** For intermediate blocks, the unpacking module $n^l$ is necessary to transform the packed, unstructured sequential node embeddings back into the structured counterparts that can be processed alternately by the local encoder of the next block. It is worth mentioning that the last block $L$ does not require an unpacking module. Instead, a readout function $e$ is defined to return the final node embeddings consumed by the task-specific decoding head:

$$\bar{\mathcal{H}}^l = n^l(\bar{H}^l), \qquad \bar{H}^L = e(\bar{H}^{L-1}) \in \mathbb{R}^{N \times D^H} \qquad (7)$$

where $\bar{\mathcal{H}}^l = \{\bar{H}_i^l\}_{i=0}^{M-1}$ is the set of node embeddings $\bar{H}_i$ for $M$ different patches, $e$ is the readout function, and $D^H$ is the dimension of the output node embeddings. The readout function is defined to be a MAX, MEAN, or LAST pooling layer.

### 3.5 Improving Over-Smoothing by Alternating Architecture

Over-smoothing is a critical problem in graph representation learning, where MPNNs fall short in encoding long-range dependencies beyond a few layers of message-passing. This issue is magnified in dynamic graphs when temporal long-range dependencies intersect with structural patterns. MPNNs typically fall into the over-smoothing regime beyond a few layers (e.g., 3), which may not be sufficient to capture long-range temporal dynamics. We propose to address this problem by letting the Transformer widen up the temporal contextual node-wise scope beyond a few hops in an alternating manner. For instance, a 3-layer MPNN encoder can reach patterns up to 9 hops away in Todyformer.

## 4 Experimental Evaluation

In this section, we evaluate the generalization performance of Todyformer through a rigorous empirical assessment spanning a wide range of benchmark datasets across downstream tasks. First, the experimental setup is explained, and a comparison with the state-of-the-art (SoTA) on dynamic graphs is provided. Next, the quantitative results are presented. Later, in-depth comparative analysis and ablation studies are provided to further highlight the role of the design choices in this work.

Table 2: Future Link Prediction performance on the test set of TGBL datasets measured using Mean Reciprocal Rank (MRR). The baseline results are directly taken from Huang et al. (2023).

| Model | Wiki | Review | Coin | Comment | Flight | Avg. Rank $\downarrow$ |
|---|---|---|---|---|---|---|
| Dyrep | $0.366 \pm 0.014$ | $0.367 \pm 0.013$ | $0.452 \pm 0.046$ | $0.289 \pm 0.033$ | $0.556 \pm 0.014$ | 4.4 |
| TGN | $0.721 \pm 0.004$ | $\mathbf{0.532 \pm 0.020}$ | $\underline{0.586 \pm 0.037}$ | $0.379 \pm 0.021$ | $0.705 \pm 0.020$ | 2 |
| CAW | $\mathbf{0.791 \pm 0.015}$ | $0.194 \pm 0.004$ | $OOM$ | $OOM$ | $OOM$ | 4.4 |
| TCL | $0.712 \pm 0.007$ | $0.200 \pm 0.010$ | $OOM$ | $OOM$ | $OOM$ | 4.8 |
| GraphMixer | $0.701 \pm 0.014$ | $\underline{0.514 \pm 0.020}$ | $OOM$ | $OOM$ | $OOM$ | 4.4 |
| EdgeBank | $0.641$ | $0.0836$ | $0.1494$ | $0.364$ | $0.580$ | 4.6 |
| **Todyformer** | $\underline{0.7738 \pm 0.004}$ | $0.5104 \pm 86e\text{-}4$ | $\mathbf{0.689 \pm 18e\text{-}4}$ | $\mathbf{0.762 \pm 98e\text{-}4}$ | $\mathbf{0.777 \pm 0.014}$ | **1.6** |

## 4.1 EXPERIMENTAL SETUP

**Baselines**: The performance of Todyformer is compared with a wide spectrum of dynamic graph encoders, ranging from random-walk based to attentive memory-based approaches: DyRep (Trivedi et al., 2019), JODIE (Kumar et al., 2019), TGAT (Xu et al., 2020), TGN (Rossi et al., 2020), CaW (Wang et al., 2021), NAT Luo & Li (2022), and DyG2Vec Alomrani et al. (2022). CAW samples temporal random walks and learns temporal motifs by counting node occurrences in each walk. NAT constructs temporal node representations using a cache that stores a limited set of historical interactions for each node. DyG2Vec introduces a window-based MPNN that attentively aggregates messages in a window of recent interactions. Recently, GraphMixer Cong et al. (2023) is presented as a simple yet effective MLP-Mixer-based dynamic graph encoder. Dygformer Yu et al. (2023) also presents a Transformer architecture that encodes the one-hop node neighborhoods.

**Downstream Tasks**: We evaluate all models on both FLP and DNC. In FLP, the goal is to predict the probability of future edges occurring given the source and destination nodes, and the timestamp. For each positive edge, we sample a negative edge that the model is trained to predict as negative. The DNC task involves predicting the ground-truth label of the source node of a future interaction. Both tasks are trained using the binary cross-entropy loss function. For FLP, we evaluate all models in both transductive and inductive settings. The latter is a more challenging setting where a model makes predictions on unseen nodes. The Average Precision (AP) and the Area Under the Curve (AUC) metrics are reported for the FLP and DNC tasks, respectively. DNC is evaluated using AUC due to the class imbalance issue.

**Datasets**: In the first set of experiments, we use 5 real-world datasets for FLP: MOOC, and LastFM (Kumar et al., 2019); SocialEvolution, Enron, and UCI (Wang et al., 2021). Three real-world datasets including Wikipedia, Reddit, MOOC (Kumar et al., 2019) are used for DNC as well. These datasets span a wide range of the number of nodes and interactions, timestamp ranges, and repetition ratios. The dataset statistics are presented in Appendix 6.1. We employ the same 70%-15%-15% chronological split for all datasets, as outlined in (Wang et al., 2021). The datasets are split differently under two settings: Transductive and Inductive. All benchmark datasets are publicly available. We follow similar experimental setups to (Alomrani et al., 2022; Wang et al., 2021) on these datasets to split them into training, validation, and test sets under the transductive and inductive settings.

In the second set of experiments, we evaluate Todyformer on the Temporal Graph Benchmark for link prediction datasets (TGBL) Huang et al. (2023). The goal is to target large-scale and real-world experimental setups with a higher number of negative samples generated based on two policies: random and historical. The deliberate inclusion of such negative edges aims to address the substantial bias inherent in negative sampling techniques, which can significantly affect model performance. Among the five datasets, three are extra-large-scale, where model training on a regular setup may take weeks of processing. We follow the experimental setups similar to Huang et al. (2023) to evaluate our model on TGBL (e.g., number of trials or negative sampling).

**Model Hyperparameters**: Todyformer has a large number of hyperparameters to investigate. There are common design choices, such as activation layers, normalization layers, and skip connections that we assumed the results are less sensitive to in order to reduce the total number of trials. We chose $L = 3$ for the number of blocks in the encoder. The GNN and Transformers have three and two layers, respectively. The neighbor sampler in the local encoder uniformly samples $(64, 1, 1)$ number of neighbors for 3 hops. The model employs uniform sampling within the window instead of selecting the latest $N$ neighbors of a node (Xu et al., 2020; Rossi et al., 2020). For the DNC task, following prior work Rossi et al. (2020), the decoder is trained on top of the frozen encoder pre-trained on FLP.

Table 3: Dynamic Node Classification performance in AUC (Mean $\pm$ Std) on the test set. Avg. Rank reports the mean rank of a method across all datasets.

| Model | Wikipedia | Reddit | MOOC | Avg. Rank $\downarrow$ |
|---|---|---|---|---|
| TGAT | $0.800 \pm 0.010$ | $\mathbf{0.664 \pm 0.009}$ | $0.673 \pm 0.006$ | 3.6 |
| JODIE | $0.843 \pm 0.003$ | $0.566 \pm 0.016$ | $0.672 \pm 0.002$ | 4.6 |
| Dyrep | $\mathbf{0.873 \pm 0.002}$ | $0.633 \pm 0.008$ | $0.661 \pm 0.012$ | 4 |
| TGN | $0.828 \pm 0.004$ | $0.655 \pm 0.009$ | $0.674 \pm 0.007$ | 3.3 |
| DyG2Vec | $0.824 \pm 0.050$ | $0.649 \pm 0.020$ | $\mathbf{0.785 \pm 0.005}$ | 3.3 |
| **Todyformer** | $0.861 \pm 0.017$ | $0.656 \pm 0.005$ | $0.745 \pm 0.009$ | **2** |

## 4.2 Experimental Results

**Future Link Prediction**: We present a comparative analysis of AP scores on the test set for future link prediction (both transductive and inductive) across several baselines in Table 1. Notably, a substantial performance gap is evident in the transductive setting, with Todyformer outperforming the second-best model by margins exceeding $1.2\%$, $1.6\%$, $0.6\%$, and $0.5\%$ on the MOOC, LastFM, UCI, and SocialEvolve datasets, respectively. Despite the large scale of the SocialEvolve dataset with around 2 million edges, our model achieves SoTA performance on this dataset. This observation reinforces the conclusions drawn in Xu et al. (2020), emphasizing the pivotal role played by recent temporal links in the future link prediction task. Within the inductive setting, Todyformer continues to exhibit superior performance across all datasets. The challenge posed by predicting links over unseen nodes impacts the overall performance of most methods. However, Todyformer consistently outperforms the baselines' results on all datasets in Table 1. These empirical results support the hypothesis that model expressiveness has significantly improved while enhancing the generalization under the two experimental settings. Additionally, Todyformer outperforms the two latest SoTA methods, namely GraphMixer Cong et al. (2023) and Dygformer Yu et al. (2023). The results further validate that dynamic graphs require encoding of long-range dependencies that cannot be simply represented by short-range one-hop neighborhoods. This verifies that multi-scale encoders like Todyformer are capable of learning inductive biases across various domains.

Additionally, the performance of Todyformer on two small and three large TGBL datasets is presented in Table 2. On extra-large TGBL datasets (Coin, Comment, and Flight), Todyformer outperforms the SoTA with significant margins, exceeding $11\%$, $39\%$, and $7\%$, respectively. This interestingly supports the hypothesis behind the expressive power of the proposed model to scale up to the data domains with extensive long-range interactions. In the case of smaller datasets like TGBL-Wiki and TGBL-Review, our approach attains the second and third positions in the ranking, respectively. It should be noted that hyperparameter search has not been exhausted during experimental evaluation. The average ranking reveals that Todyformer is ranked first, followed by TGN in the second place in this challenging experimental setup.

**Dynamic Node classification**: Todyformer has undergone extensive evaluation across three datasets dedicated to node classification. In these datasets, dynamic sparse labels are associated with nodes within a defined time horizon after interactions. This particular task grapples with a substantial imbalanced classification challenge. Table 3 presents the AUC metric, known for its robustness toward class imbalance, across various methods on the three datasets. Notably, Todyformer demonstrates remarkable performance, trailing the best by only $4\%$ on the MOOC dataset and $1\%$ on both the Reddit and Wikipedia datasets. Across all datasets, Todyformer consistently secures the second-best position. However, it is important to acknowledge that no model exhibits consistent improvement across all datasets, primarily due to the presence of data imbalance issues inherent in anomaly detection tasks Ranshous et al. (2015). To establish the ultimate best model, we have computed the average ranks of various methods. Todyformer emerges as the top performer with an impressive rank of 2, validating the overall performance improvement.

## 4.3 Ablation Studies and sensitivity analysis

We conducted an evaluation of the model's performance across various parameters and datasets to assess the sensitivity of the major hyperparameters. Figure 3 illustrates the sensitivity analysis regarding the window size and the number of patches, with one parameter remaining constant while the other changes. As highlighted in Xu et al. (2020), recent and frequent interactions dis-

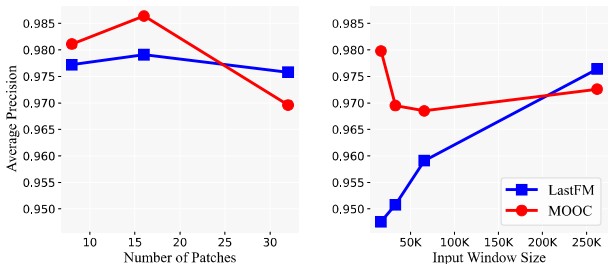

| Dataset | G. E. | P. E. | Alt. 3 | AP |
|---|---|---|---|---|
| MOOC | ✗ | ✗ | ✗ | 0.980 |
| | ✓ | ✗ | ✗ | 0.981 |
| | ✓ | ✓ | ✗ | 0.987 |
| | ✓ | ✓ | ✓ | **0.992** |
| LastFM | ✗ | ✗ | ✗ | 0.960 |
| | ✓ | ✗ | ✗ | 0.961 |
| | ✓ | ✓ | ✗ | 0.965 |
| | ✓ | ✓ | ✓ | **0.976** |
| UCI | ✗ | ✗ | ✗ | 0.981 |
| | ✓ | ✗ | ✗ | 0.983 |
| | ✓ | ✓ | ✗ | 0.987 |
| | ✓ | ✓ | ✓ | **0.993** |
| SocialEvolution | ✗ | ✗ | ✗ | 0.987 |
| | ✓ | ✗ | ✗ | 0.987 |
| | ✓ | ✓ | ✗ | 0.989 |
| | ✓ | ✓ | ✓ | **0.991** |

Figure 3: Sensitivity analysis on the number of patches and input window size values on MOOC and LastFM. The plot on the left has a fixed input window size of 262,144, while the one on the right has 32 patches.

Table 4: Ablation studies on three major components: global encoder (G. E.), Positional Encoding (P. E.), and number of alternating blocks (Alt. 3)

play enhanced predictability of future interactions. This predictability is particularly advantageous for datasets with extensive long-range dependencies, favoring the utilization of larger window size values to capture recurrent patterns. Conversely, in datasets where recent critical interactions reflect importance, excessive emphasis on irrelevant information becomes prominent when employing larger window sizes. Our model, complemented by uniform neighbor sampling, achieves a balanced equilibrium between these contrasting sides of the spectrum. As an example, the right plot in Figure 3 demonstrates that with a fixed number of patches (i.e., 32), an increase in window size leads to a corresponding increase in the validation AP metric on the LastFM dataset. This trend is particularly notable in LastFM, which exhibits pronounced long-range dependencies, in contrast to datasets like MOOC and UCI with medium- to short-range dependencies.

In contrast, in Figure 3 on the left side, with a window size of 262k, we vary the number of patches. Specifically, for the MOOC dataset, performance exhibits an upward trajectory with an increase in the number of patches from 8 to 16; however, it experiences a pronounced decline when the number of patches reaches 32. This observation aligns with the inherent nature of MOOC datasets, characterized by their relatively high density and reduced prevalence of long-range dependencies. Conversely, when considering LastFM data, the model maintains consistently high performance even with 32 patches. In essence, this phenomenon underscores the model's resilience on datasets featuring extensive long-range dependencies, illustrating a trade-off between encoding local and contextual features by tweaking the number of patches.

In Table 4, we conducted ablation studies on the major design choices of the encoding network to assess the roles of the three hyperparameters separately: a) Global encoder, b) Alternating mode, and c) Positional Encoding. Across the four datasets, the alternating approach exhibits significant performance variation compared to others, ensuring the mitigation of over-smoothing and the capturing of long-range dependencies. The outcomes of the single-layer vanilla transformer as a global encoder attain the second-best position, affirming the efficacy of our global encoder in enhancing expressiveness. Finally, the global encoder without PE closely resembles the model with only a local encoder (i.e., DyG2Vec MPNN model).

# 5 CONCLUSION

We propose Todyformer, a tokenized graph Transformer for dynamic graphs, where over-smoothing and over-squashing are empirically improved through a local and global encoding architecture. We present how to adapt the best practices of Transformers in various data domains (e.g., Computer Vision) to dynamic graphs in a principled manner. The primary novel components are patch generation, structure-aware tokenization using typical MPNNs that locally encode neighborhoods, and the utilization of Transformers to aggregate global context in an alternating fashion. The consistent experimental gains across different experimental settings empirically support the hypothesis that the SoTA dynamic graph encoders severely suffer from over-squashing and over-smoothing phenomena, especially on the real-world large-scale datasets introduced in TGBL. We hope Todyformer sheds light on the underlying aspects of dynamic graphs and opens up the door for further principled investigations on dynamic graph transformers.

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
