# TODYFORMER: TOWARDS HOLISTIC DYNAMIC GRAPH TRANSFORMERS WITH STRUCTURE-AWARE TOKENIZATION

## A APPENDIX

### A.1 DATASET STATISTICS

In this section, we provide an overview of the statistics pertaining to two distinct sets of datasets utilized for the tasks of Future Link Prediction (FLP) and Dynamic Node Classification (DNC). The initial set, detailed in Table 1, presents information regarding the number of nodes, edges, and unique edges across seven datasets featured in Table 10 and Table 12. For these three datasets, namely Reddit, Wikipedia, and MOOC, all edge features have been incorporated, where applicable. Furthermore, within this table, the last column represents the percentage of Repetitive Edges, which signifies the proportion of edges that occur more than once within the dynamic graph.

Table 1: Dynamic Graph Datasets. **% Repetitive Edges**: % of edges which appear more than once in the dynamic graph.

| Dataset | # Nodes | # Edges | # Unique Edges | Edge Features | Node Labels | Bipartite | % Repetitive Edges |
|---|---|---|---|---|---|---|---|
| Reddit | 11,000 | 672,447 | 78,516 | ✓ | ✓ | ✓ | 54% |
| Wikipedia | 9,227 | 157,474 | 18,257 | ✓ | ✓ | ✓ | 48% |
| MOOC | 7,144 | 411,749 | 178,443 | ✓ | ✓ | ✓ | 53% |
| LastFM | 1980 | 1,293,103 | 154,993 | | | ✓ | 68% |
| UCI | 1899 | 59,835 | 13838 | | | ✓ | 62% |
| Enron | 184 | 125,235 | 2215 | | | | 92% |
| SocialEvolution | 74 | 2,099,519 | 2506 | | | | 97% |

#### A.1.1 TGB DATASET

In this section, we present the characteristics of datasets as proposed by the Dynamic Graph Encoder Leaderboard Huang et al. (2023). Similar to previous benchmark datasets, we have conducted comparisons regarding the number of nodes, edges, and type of graphs. Additionally, we report the Number of Steps and the Surprise Index, as defined in Poursafaei et al. (2022), which illustrates the ratio of test edges that were not observed during the training phase.

Table 2: Statistics of TGBL Dynamic Graph Datasets

| Dataset | # Nodes | # Edges | # Steps | Edge Features | Bipartite | Surprise Index Poursafaei et al. (2022) |
|---|---|---|---|---|---|---|
| Wiki | 9,227 | 157,474 | 152,757 | ✓ | ✓ | 0.108 |
| Review | 352,637 | 4,873,540 | 6,865 | ✓ | ✓ | 0.987 |
| Coin | 638,486 | 22,809,486 | 1,295,720 | ✓ | | 0.120 |
| Comment | 994,790 | 44,314,507 | 30,998,030 | ✓ | | 0.823 |
| Flight | 18143 | 67,169,570 | 1,385 | ✓ | | 0.024 |

### A.2 IMPLEMENTATION DETAILS

In this section, we elucidate the intricacies of our implementation, providing a comprehensive overview of the specific parameters our model accommodates during hyperparameter optimization. Subsequently, we delve into a discussion of the optimal configurations and setups that yield the best performance for our proposed architecture.

Furthermore, in addition to an in-depth discussion of the baselines incorporated into our paper, we also offer a comprehensive overview of the respective hyperparameter configurations in this section. We are confident that with the open-sourcing of our code upon acceptance and the thorough descriptions of our model and baseline methodologies presented in the paper, our work is fully reproducible.

### A.2.1 EVALUATION PROTOCOL

**Transductive Setup:** Under the transductive setting, a dataset is split normally by time, i.e., the model is trained on the first $70\%$ of links, validated on $\%15$ and tested on the rest.

**Inductive Setup:** In the inductive setting, we strive to test the model's prediction performance on edges with unseen nodes. Therefore, following (Wang et al., 2021), we randomly assign $10\%$ of the nodes to the validation and test sets and remove any interactions involving them in the training set. Additionally, to ensure an inductive setting, we remove any interactions not involving these nodes from the test set.

### A.2.2 LOSS FUNCTION

As previously discussed in the main body of this paper, we focus on two specific downstream tasks: Future Link Prediction (FLP) and Dynamic Node Classification (DNC). For the former, we employ the binary cross-entropy loss, while for the latter, our model is trained through the minimization of the cross-entropy loss function. The formula for the binary cross-entropy loss is presented below:

$$H(y, \hat{y}) = -(y \cdot \log(\hat{y}) + (1 - y) \cdot \log(1 - \hat{y})) \tag{1}$$

where $y \in \{0, 1\}$ is the true label, and $\hat{y}$ is the predicted probability that the instance belongs to class 1. Moreover, the formulation of the cross-entropy loss is as follows:

$$H(y, \hat{y}) = -\sum_i y_i \cdot \log(\hat{y}_i) \tag{2}$$

where $i$ represents the index over all classes, $y_i$ is the true probability of the sample belonging to class $i$, encoded as a one-hot vector. It is 1 for the true class and 0 for all other classes. Finally, $\hat{y}_i$ is the predicted probability of the sample belonging to class $i$.

### A.2.3 BEST HYPERPARAMETERS FOR BENCHMARK DATASETS.

Table 3 displays the hyperparameters that have been subjected to experimentation and tuning for each dataset. For each parameter, a range of values has been tested as follows:

- Window Size (W): This parameter signifies the window length chosen for selecting the input subgraph based on edge timestamps. It falls within the range of $\in \{$ 16384, 32768 ,65536, 262144 $\}$.

- Number of Patches: This parameter indicates the count of equal and non-overlapping chunks for each input subgraph. It is the range of $\in \{8, 16, 32\}$.

- #Local Encoders: This parameter represents the number of local encoder layers within each block, and its value falls within the range of $\in \{1, 2\}$.

- Neighbor Sampling (NS) mode: $\in \{uniform, last\}$. In the case of a uniform Neighbor Sampler (NS), it uniformly selects samples from the 1-hop interactions of a given node. Conversely, in last mode, it samples from the most recent interactions.

- Anchor Node Mode: $\in \{GlobalTarget, LocalInput, LocalTarget\}$ depending on the mechanism of neighbor sampling we can sample from nodes within all patches (LocalInput), nodes within the next patch (LocalTarget), or global target nodes (GlobalTarget).

- Batch Size: $\in \{8, 16, 32, 64\}$

- Positional Encoding: $\in \{SineCosine, Time2Vec, Identity, Linear\}$

| Dataset | Window Size ($W$) | Number of Patches | #Local Encoders | NS Mode | Anchor Node Mode | Batch Size |
|---|---|---|---|---|---|---|
| Reddit | 262144 | 32 | 2 | uniform | GlobalTarget | 8 |
| Wikipedia | 65536 | 8 | 2 | uniform | GlobalTarget | 8 |
| MOOC | 65536 | 8 | 2 | uniform | GlobalTarget | 8 |
| LastFM | 262144 | 32 | 2 | uniform | GlobalTarget | 8 |
| UCI | 65536 | 8 | 2 | uniform | GlobalTarget | 8 |
| Enron | 65536 | 8 | 2 | uniform | GlobalTarget | 8 |
| SocialEvolution | 65536 | 8 | 2 | uniform | GlobalTarget | 8 |

Table 3: Best Parameters of the model pipeline after Hyperparameter search

SineCosine is utilized as the Positional Encoding (PE) method following the experiments conducted in Appendix A.3.3.

**Selecting Best Checkpoint:** Throughout all experiments, the models undergo training for a duration of 100 epochs, with the best checkpoints selected for testing based on their validation Average Precision (AP) performance.

### A.2.4 Best Hyperparameters for TGBL dataset

In this section, we present the optimal hyperparameters used in our architecture design for each TGBL dataset. We conducted hyperparameter tuning for all TGBL datasets; however, due to time constraints, we explored a more limited set of parameters for the large-scale dataset. Despite Todyformer outperforming its counterparts on these datasets, there remains potential for further improvement through an extensive hyperparameter search.

| Dataset | Window Size ($W$) | Number of Patches | First-hop NS size | NS Mode | Anchor Node Mode | Batch Size |
|---|---|---|---|---|---|---|
| TGBWiki | 262144 | 32 | 256 | uniform | GlobalTarget | 32 |
| TGBReview | 262144 | 32 | 64 | uniform | GlobalTarget | 64 |
| TGBComment | 65536 | 8 | 64 | uniform | GlobalTarget | 256 |
| TGBCOin | 65536 | 8 | 64 | uniform | GlobalTarget | 96 |
| TGBFlight | 65536 | 8 | 64 | uniform | GlobalTarget | 128 |

Table 4: Optimal Window size $W$ for downstream training.

### A.3 More Experimental Result

In this section, we present the additional experiments conducted and provide an analysis of the derived results and conclusions.

### A.3.1 FLP result on Benchmark Datasets

Table 5 is an extension of Table 10, now incorporating the Wikipedia and Reddit datasets. Notably, for these two datasets, Todyformer attains the highest test Average Precision (AP) score in the Transductive setup. However, it secures the second-best and third-best positions in the Inductive setup for these Wikipedia and Reddit respectively. While the model does not attain the top position on these two datasets for inductive setup, its performance is only marginally below that of state-of-the-art (SOTA) models, which have previously achieved accuracy levels exceeding 99% Average Precision (AP).

### A.3.2 FLP validation result on TGBL dataset

As discussed in the paper, Todyformer has been compared to baseline methods using the TGBL dataset. Table 6 represents an extension of Table 11 specifically for validation (MRR). The results presented in both tables are in line with counterpart methods outlined in the paper by Huang et al. (2023). It is important to note that for the larger datasets, TCL, GraphMIxer, and EdgeBank were found to be impractical due to memory constraints, as mentioned in the paper.

Table 5: Future link Prediction Performance in AP (Mean $\pm$ Std). **Bold** font and ul font represent first- and second-best performance respectively.

| Setting | Model | Wikipedia | Reddit | MOOC | LastFM | Enron | UCI | SocialEvol. |
|---|---|---|---|---|---|---|---|---|
| Transductive | JODIE | $0.956 \pm 0.002$ | $0.979 \pm 0.001$ | $0.797 \pm 0.01$ | $0.691 \pm 0.010$ | $0.785 \pm 0.020$ | $0.869 \pm 0.010$ | $0.847 \pm 0.014$ |
| | DyRep | $0.955 \pm 0.004$ | $0.981 \pm 1e\text{-}4$ | $0.840 \pm 0.004$ | $0.683 \pm 0.033$ | $0.795 \pm 0.042$ | $0.524 \pm 0.076$ | $0.885 \pm 0.004$ |
| | TGAT | $0.968 \pm 0.001$ | $0.986 \pm 3e\text{-}4$ | $0.793 \pm 0.006$ | $0.633 \pm 0.002$ | $0.637 \pm 0.002$ | $0.835 \pm 0.003$ | $0.631 \pm 0.001$ |
| | TGN | $0.986 \pm 0.001$ | $0.985 \pm 0.001$ | $0.911 \pm 0.010$ | $0.743 \pm 0.030$ | $0.866 \pm 0.006$ | $0.843 \pm 0.090$ | $0.966 \pm 0.001$ |
| | CaW | $0.976 \pm 0.007$ | $0.988 \pm 2e\text{-}4$ | $0.940 \pm 0.014$ | $0.903 \pm 1e\text{-}4$ | $0.970 \pm 0.001$ | $0.939 \pm 0.008$ | $0.947 \pm 1e\text{-}4$ |
| | NAT | $0.987 \pm 0.001$ | $0.991 \pm 0.001$ | $0.874 \pm 0.004$ | $0.859 \pm 1e\text{-}4$ | $0.924 \pm 0.001$ | $0.944 \pm 0.002$ | $0.944 \pm 0.010$ |
| | GraphMixer | $0.974 \pm 0.001$ | $0.975 \pm 0.001$ | $0.835 \pm 0.001$ | $0.862 \pm 0.003$ | $0.824 \pm 0.001$ | $0.932 \pm 0.006$ | $0.935 \pm 3e\text{-}4$ |
| | Dygformer | $0.991 \pm 0.0001$ | $0.992 \pm 0.0001$ | $0.892 \pm 0.005$ | $0.901 \pm 0.003$ | $0.926 \pm 0.001$ | $0.959 \pm 0.001$ | $0.952 \pm 2e\text{-}4$ |
| | DyG2Vec | $0.995 \pm 0.003$ | $0.996 \pm 2e\text{-}4$ | $0.980 \pm 0.002$ | $0.960 \pm 1e\text{-}4$ | $0.991 \pm 0.001$ | $0.988 \pm 0.007$ | $0.987 \pm 2e\text{-}4$ |
| | **Todyformer** | **$0.996 \pm 2e\text{-}4$** | **$0.998 \pm 8e\text{-}5$** | **$0.992 \pm 7e\text{-}4$** | **$0.976 \pm 3e\text{-}4$** | **$0.995 \pm 6e\text{-}4$** | **$0.994 \pm 4e\text{-}4$** | **$0.992 \pm 1e\text{-}4$** |
| Inductive | JODIE | $0.891 \pm 0.014$ | $0.865 \pm 0.021$ | $0.707 \pm 0.029$ | $0.865 \pm 0.03$ | $0.747 \pm 0.041$ | $0.753 \pm 0.011$ | $0.791 \pm 0.031$ |
| | DyRep | $0.890 \pm 0.002$ | $0.921 \pm 0.003$ | $0.723 \pm 0.009$ | $0.869 \pm 0.015$ | $0.666 \pm 0.059$ | $0.437 \pm 0.021$ | $0.904 \pm 3e\text{-}4$ |
| | TGAT | $0.954 \pm 0.001$ | $0.979 \pm 0.001$ | $0.805 \pm 0.006$ | $0.644 \pm 0.002$ | $0.693 \pm 0.004$ | $0.820 \pm 0.005$ | $0.632 \pm 0.005$ |
| | TGN | $0.974 \pm 0.001$ | $0.954 \pm 0.002$ | $0.855 \pm 0.014$ | $0.789 \pm 0.050$ | $0.746 \pm 0.013$ | $0.791 \pm 0.057$ | $0.904 \pm 0.023$ |
| | CaW | $0.977 \pm 0.006$ | $0.984 \pm 2e\text{-}4$ | $0.933 \pm 0.014$ | $0.890 \pm 0.001$ | $0.962 \pm 0.001$ | $0.931 \pm 0.002$ | $0.950 \pm 1e\text{-}4$ |
| | NAT | $0.986 \pm 0.001$ | $0.986 \pm 0.002$ | $0.832 \pm 1e\text{-}4$ | $0.878 \pm 0.003$ | $0.949 \pm 0.010$ | $0.926 \pm 0.010$ | $0.952 \pm 0.006$ |
| | GraphMixer | $0.966 \pm 2e\text{-}4$ | $0.952 \pm 2e\text{-}4$ | $0.814 \pm 0.002$ | $0.821 \pm 0.004$ | $0.758 \pm 0.004$ | $0.911 \pm 0.004$ | $0.918 \pm 6e\text{-}4$ |
| | Dygformer | $0.985 \pm 3e\text{-}4$ | $0.988 \pm 2e\text{-}4$ | $0.869 \pm 0.004$ | $0.942 \pm 9e\text{-}4$ | $0.897 \pm 0.003$ | $0.945 \pm 0.001$ | $0.931 \pm 4e\text{-}4$ |
| | DyG2Vec | **$0.992 \pm 0.001$** | **$0.991 \pm 0.002$** | $0.938 \pm 0.010$ | $0.979 \pm 0.006$ | $0.987 \pm 0.004$ | $0.976 \pm 0.002$ | $0.978 \pm 0.010$ |
| | **Todyformer** | $0.989 \pm 6e\text{-}4$ | $0.983 \pm 0.002$ | **$0.948 \pm 0.009$** | **$0.981 \pm 0.005$** | **$0.989 \pm 8e\text{-}4$** | **$0.983 \pm 0.002$** | **$0.9821 \pm 0.005$** |

Table 6: (Validation) Future Link Prediction performance in Validation MRR on TGB Leaderboard datasets.

| Model | TGBWiki | TGBReview | TGBCoin | TGBComment | TGBFlight | Avg. Rank ↓ |
|---|---|---|---|---|---|---|
| Dyrep | $0.411 \pm 0.015$ | $0.356 \pm 0.016$ | $0.512 \pm 0.014$ | $0.291 \pm 0.028$ | $0.573 \pm 0.013$ | 4.2 |
| TGN | $0.737 \pm 0.004$ | **$0.465 \pm 0.010$** | $0.607 \pm 0.014$ | $0.356 \pm 0.019$ | $0.731 \pm 0.010$ | 2.2 |
| CAWN | $0.794 \pm 0.014$ | $0.201 \pm 0.002$ | $OOM$ | $OOM$ | $OOM$ | 3 |
| TCL | $0.734 \pm 0.007$ | $0.194 \pm 0.012$ | $OOM$ | $OOM$ | $OOM$ | 5 |
| GraphMixer | $0.707 \pm 0.014$ | $0.411 \pm 0.025$ | $OOM$ | $OOM$ | $OOM$ | 4 |
| EdgeBank | $0.641$ | $0.0894$ | $0.1244$ | $0.388$ | $0.492$ | 4.6 |
| **Todyformer** | **$0.799 \pm 0.0092$** | $0.4321 \pm 0.0040$ | **$0.6852 \pm 0.0021$** | **$0.7402 \pm 0.0037$** | **$0.7932 \pm 0.014$** | **1.2** |

### A.3.3 COMPLEMENTARY SENSITIVITY ANALYSIS AND ABLATION STUDY

In this section, we have presented the specifics of sensitivity and ablation experiments, which, while of lesser significance in our hyper-tuning mechanism, contribute valuable insights. In all tables, the Average Precision scores reported in the table are extracted from the same epoch on the validation set. Table 7 showcases the influence of varying input window sizes and patch sizes on two datasets. Table 8 illustrates the effects of various PEs, including SineCosine, Time2VecKazemi et al. (2019), Identity, Linear, and a configuration utilizing Local Input as the Anchor Node Mode. The table presents a comparison of results for these different PEs. Notably, the architecture appears to be relatively insensitive to the type of PE used, as the results all fall within a similar range. However, it is worth mentioning that SineCosine PE slightly outperforms the others. Consequently, SineCosine PE will be selected as the primary module for all subsequent experiments.

In Table9, an additional ablation study has been conducted to elucidate the influence of positions tagged to each node before being input to the Positional Encoder module. Various mechanisms for adding positions are delineated as follows:

- Without PE: No position is utilized or tagged to the nodes.

- Random Index: An index is randomly generated and added to the embeddings of a given node.

- Patch Index: The index of the patch from which the embedding of the given node originates is used as a position.

- Edge Time: The most recent edge time within its patch is employed as a position.

- Edge Index: The index of the most recent interaction within the corresponding patch is utilized as a position.

As evident from the findings in this table, the validation performance exhibits high sensitivity to the positional encoder's outcomes. Specifically, the model without positional encoder (PE) and the model with random indices manifest the lowest performance among all available options. Consistent

with our expectations from previous experiments, the patch index yields the highest performance, providing a compelling rationale for its incorporation into the architecture.

Table 7: Sensitivity analysis on number of patches and target window size

| dataset | Input Window size | Number of Patches | Average Precision ↑ |
|---------|-------------------|-------------------|---------------------|
| LastFM  | 262144            | 8                 | 0.9772              |
| LastFM  | 262144            | 16                | 0.9791              |
| LastFM  | 262144            | 32                | 0.9758              |
| MOOC    | 262144            | 8                 | 0.9811              |
| MOOC    | 262144            | 16                | 0.9864              |
| MOOC    | 262144            | 32                | 0.9696              |
| LastFM  | 16384             | 32                | 0.9476              |
| LastFM  | 32768             | 32                | 0.9508              |
| LastFM  | 65536             | 32                | 0.9591              |
| LastFM  | 262144            | 32                | 0.9764              |
| MOOC    | 16384             | 32                | 0.9798              |
| MOOC    | 32768             | 32                | 0.9695              |
| MOOC    | 65536             | 32                | 0.9685              |
| MOOC    | 262144            | 32                | 0.9726              |

## A.4 COMPUTATIONAL COMPLEXITY

### A.4.1 QUALITATIVE ANALYSIS FOR TIME AND MEMORY COMPLEXITIES

In this section, we delve into the detailed measurement and discussion of the computational complexity of Todyformer. Initially, we adopt the assumption that the time complexity of Transformers is $O(X^2)$ for an input sequence of length $X$. The primary complexity of Todyformer encompasses both the complexity of the Message Passing Neural Network (MPNN) component and the complexity of the Transformer. To elaborate further, assuming we have a sparse dynamic graph with temporal attributes, we can replace the complexity of MPNNs with $O(l \times (N + E))$, where N and E represent the number of nodes and edges within the temporal input subgraph, and $l$ is the number of MPNN layers for the Graph Neural Network (GNN) tokenizer. In the transformer part, $N$ unique nodes are fed into the Multihead-Attention module. If the maximum length of the sequence fed to the Transformer is $N_a$, then the complexity of the Multihead-Attention module is $O(N_a^2)$. Notably, $N_a$ is at most equal to $M$, the number of patches. This scenario occurs when a node appears in all M patches and has interactions in all patches. Consequently, if $L$ is the number of blocks the final complexity is given by:

$$O(L \times l \times (N + E) + L \times N \times M^2) \approx O(N + E) \qquad (3)$$

The LHS part of Equation 3 can be simplified to RHS if we assume that $L$, $l$, and $M^2$ are negligible compared to $N$ and $E$. The RHS of this equation is the time complexity of GNNs for sparse graphs.

### A.4.2 TRAINING/INFERENCE SPEED

In this section, an analysis of Figure 1 is provided, depicting the performance versus inference time across three sizable datasets. Considering the delicate trade-off between performance and complexity, our models surpass all others in terms of Average Precision (AP) while concurrently

| Positional Encoding | Anchor_Node_Mode | Average Precision ↑ |
|---------------------|------------------|---------------------|
| SineCosinePos       | global target    | 0.9901              |
| Time2VecPos         | global target    | 0.989               |
| IdentityPos         | global target    | 0.99                |
| LinearPos           | global target    | 0.9886              |
| SineCosinePos       | local input      | 0.9448              |

Table 8: **Ablation Study on Positional Encoding Options on MOOC Dataset:** This table compares the validation performance at the same epoch across various setups.

| Positional Encoding (PE) Input | Average Precision ↑ |
|---|---|
| without PE | 0.9872 |
| random index | 0.9873 |
| patch index | 0.9889 |
| edge time | 0.9886 |
| edge index | 0.9877 |

Table 9: **Ablation Study on the Input of Positional Encoding on MOOC Dataset:** This table compares the validation performance at the same epoch across various types of positions tagged to nodes before PE layer.

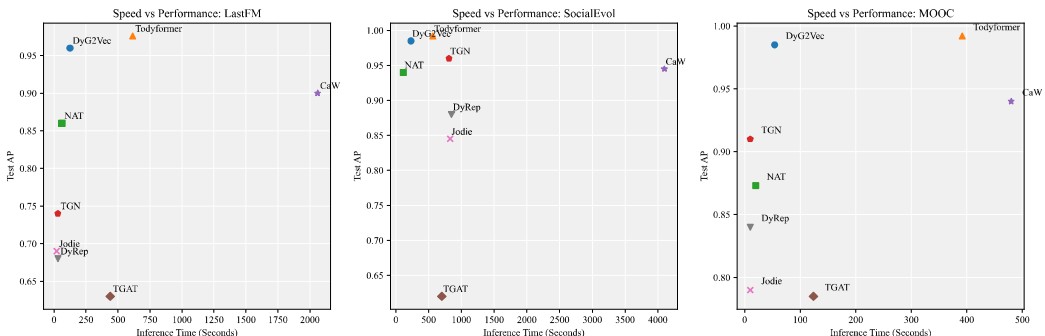

Figure 1: The performance versus inference time across LastFM, SocialEvol and MOOC datasets

positioning in the left segment of the diagrams, denoting the lowest inference time. Notably, as depicted in Figure 4, Todyformer remains lighter and less complex than state-of-the-art (SOTA) models like CAW across all datasets.

## A.5  DISCUSSION ON OVER-SMOOTHING AND OVER-SQUASHING

In Figure 2, the blue curve illustrates the Average Precision performance of dynamic graph Message Passing Neural Networks (MPNNs) across varying numbers of layers. Notably, an observed trend indicates that as the number of layers increases, the performance experiences a decline—a characteristic manifestation of oversmoothing and oversquashing phenomena.

Within the same figure, the red square dots represent the performance of MPNNs augmented with transformers, specifically Todyformer with a single block. It is noteworthy that the increase in the number of MPNN layers from 3 to 9 in this configuration results in a comparatively minor performance drop compared to traditional MPNNs.

Furthermore, the yellow circles denote the performance of Todyformer with an alternating mode, where the total number of MPNNs is 9, and three blocks are incorporated. In this setup, a transformer is introduced after every 3 MPNN layers. Strikingly, this configuration outperforms all others, especially those that stack a similar number of MPNN layers without the insertion of a transformer layer in the middle of the architecture. This empirical observation serves as a significant study, highlighting the efficacy of our architecture in addressing oversmoothing and oversquashing challenges.

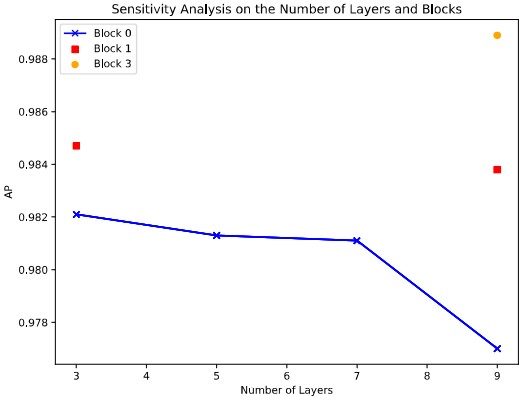

Figure 2: Sensitivity Analysis on the Number of Layers and Blocks on Mooc Dataset