# OpenReview forum: "Todyformer: Towards Holistic Dynamic Graph Transformers with Structure-Aware Tokenization"
_ICLR.cc/2024/Conference — Submitted to ICLR 2024_

### Official Review · Reviewer_787D · 2023-10-25

**Soundness:** 2 fair
**Presentation:** 1 poor
**Contribution:** 1 poor
**Rating:** 3
**Confidence:** 4

**Summary:**

This paper proposes Todyformer, a novel Transformer-based neural network for dynamic graphs, to address the problems of over-smoothing/squashing caused by Dynamic Message-Passing Neural Networks and learning long-range dependencies. The experiments of future link prediction and node classification are conducted to verify its effectiveness.

**Strengths:**

1.	A novel Transformer-based neural network is proposed for dynamic graph representation learning.
2.	The proposed TodyFormer achieved the best performance over 9 baselines on both transductive and inductive link prediction experiments.

**Weaknesses:**

1.	To the best of my knowledge, there is no study pointing out existing well-known temporal graph neural networks (TGNN) have the problem of over-smoothing. I also do not see any theoretical or experimental evidence on the over-smoothing problem of TGNN. Therefore, the motivation of this paper may be not solid.
2.	There are some existing works studying leveraging Transformer for dynamic graph learning, e.g., APAN [1], DygFormer [2]. What are the advantages of the proposed Todyformer over these methods?
3.	The inputs of TodyFormer are edges within a temporal window. How to set the size of this window? Does it mean that the edges in the previous time will see the edges in the latter time in the window (information leakage)? How do you prevent the information leakage problem?
4.	There are a lot of symbols used in the equations without detailed explanation, which makes it really hard to understand. For example, what is P, c in Eq. (4) ? what is positional encoding P in Eq. (5) ? what is n, e in Eq. (7) ?
5.	From ablation study (Table 4), there is really slight difference after removing modules of TodyFormer? Even if all the modules are removed, TodyFormer still has very high performance (e.g., 0.987 on social evolution). I do not understand which modules contributes the such high performance? In addition, more ablation study on other modules should be studied, e.g. (replace node2vec encoding with others).
6.	More sensitivity analysis on other hyper-parameters should be conducted.
7.	There is no training/testing time comparison with other baselines.
8.	There are many spelling and grammar and Latex errors in this paper. Please check the whole paper carefully.

**Questions:**

1.	In section 3.2, how do you partition the input graph into M non-overlapping subgraphs?
2.	In Eq. (3), what is s^l? Why H^l is the combination M node embeddings? There seems a contradiction as the author stated M are the number of subgraphs stated in previous section.
3.	In Figure 3, the results of left and right subgraphs seem contradict. On the left, the window size of LastFM is 144, and the AP score is larger than 0.975. On the right, when windows size of LastFM is less than 50k, it seems the AP score is less than 0.95. Why is that? Besides, as my comment 3, such large window size may cause severe information leakage.
4.	It is really wired that on 4 of 5 datasets, TodyFormer has the AP score over 0.99 (Table 1).

---

> ### Author Response · Authors · 2023-11-23
> **Response Part 1**
>
> > Weakness 1
>
> In the realm of continuous-time dynamic graphs, the structure resembles that of regular graphs, but with the added dimension of temporal edge features. Notably, over-smoothing becomes a pronounced issue in scenarios involving temporal edges, where any pair of nodes can engage in multiple interactions over time. It is a nontrivial task to assert that the challenges associated with over-smoothing in non-temporal graphs persist in temporal graphs. The primary focus of this work, however, is not to delve into a theoretical or empirical investigation of over-smoothing in temporal graphs. Instead, the aim is to propose strategies for enhancing model performance in the context of temporal graphs. Our empirical findings lend support to the hypothesis that this improvement is closely tied to mitigating the effects of over-smoothing and over-squashing phenomena.
>
> > Weakness 2
>
> Dygformer operates by leveraging the 1-hop neighborhood of each node to undergo patchification and encoding through Transformers . In contrast, Todyformer employs a distinctive methodology wherein the entire input graph is decomposed into multiple patches (subgraphs). Subsequently, localized tokenization utilizing Graph Neural Networks (GNNs) is applied, followed by the feeding of node embeddings across temporal dimensions into Transformers. Todyformer strategically employs GNNs to facilitate structure-aware tokenization, with the Transformer subsequently being applied to tokens originating from anchor nodes across different temporal instances. This architectural distinction contributes to the heightened expressiveness of Todyformer in comparison to Dygformer, a proposition substantiated by empirical results.
>
> > Weakness 3
>
> There is no concern regarding information leakage in the link prediction task. In this context, for a target edge at a specific time, all preceding edges until that time are considered as the historical context for the target (anchor) edge. As a result, there is no inadvertent data leakage. The model outputs node embeddings for the source and destination nodes in the target edges based on their historical context. The historical context can be constrained to a recent time window, which determines the size of the input graph received by the model. This input graph is then partitioned into equally-sized subgraphs to form patches. Subsequently, each patch is locally encoded by a message-passing Graph Neural Network (GNN) tailored for dynamic graphs. These subgraphs are then globally encoded at the node level using Transformers, as detailed in the paper.
>
> > Weakness 4
>
> Thank you for your thoughtful consideration. The errors mentioned in your feedback have been identified and rectified in the revised version of the main paper. The changes have been color coded by blue.
>
> > Weakness 6
>
> Additional sensitivity analyses on positional encoding, as well as the number of layers and blocks, have been incorporated into the supplementary materials in Appendix A.5 and A.3.3.
>
> > Weakness 7
>
> The qualitative and quantitative comparison of time complexity has been appended to Appendix A4. In A.4.1, an in-depth exploration of the model's time complexity concerning the number of nodes, edges, and patches is presented. In A.4.2, an analysis of Figure 4 is provided, depicting the performance versus inference time across three sizable datasets. Considering the delicate trade-off between performance and complexity, our models surpass all others in terms of Average Precision (AP) while concurrently positioning in the left segment of the diagrams, denoting the lowest inference time. Notably, as depicted in Figure 4, Todyformer remains lighter and less complex than state-of-the-art (SOTA) models like CAW across all datasets.
>
> > Weakness 8
>
> Thank you for your thoughtful consideration. The errors mentioned in your feedback have been identified and rectified in the revised version of the main paper. The changes have been color coded by blue.

---

### Official Review · Reviewer_dSfv · 2023-10-30

**Soundness:** 2 fair
**Presentation:** 2 fair
**Contribution:** 2 fair
**Rating:** 5
**Confidence:** 3

**Summary:**

This paper proposes a Transformer-based architecture for dynamic graphs, namely Todyformer. Experiments demonstrate that Todyformer outperforms the state-of-the-art methods on some datasets.

**Strengths:**

1. This paper is easy to follow.

**Weaknesses:**

1. The authors claim that Transformers demonstrate exceptional computational capacity, compared with Graph Neural Networks. However, the computational complexity of GNNs and traditional Transformers is $O(N)$ [1]  and $O(N^2)$ [2] respectively, where $N$ is the number of nodes in the input graph. I suggest reporting the complexity and runtime in experiments.
2. Todyformer is similar to Vision Transformer, while the authors do not provide the necessary analysis in terms of graph learning. Some suggestions are as follows.
	1. The authors may want to analyze the expressiveness of Todyformer in terms of sub-structure identification and the Weisfeiler-Leman (WL) test.
	2. The authors may want to analyze how and why Todyformer alleviates over-squashing and over-smoothing.
3. Please explain why the baseline results in Table 2 are different from those from Table 3 in [3].
4. The authors claim that the significantly low performance of Todyformer on the Wiki and Review datasets is due to the insufficient hyperparameter search. However, in my opinion, hyperparameter tuning is difficult to improve 3% accuracy.
5. Please report the standard deviation in Table 2 following the baselines. Moreover, I suggest reporting a statistically significant difference.



[1] Recipe for a General, Powerful, Scalable Graph Transformer. NeurIPS 2022.

[2] NodeFormer: A Scalable Graph Structure Learning Transformer for Node Classification. NeurIPS 2022.

[3] Temporal Graph Benchmark for Machine Learning on Temporal Graphs https://arxiv.org/pdf/2307.01026.pdf

**Questions:**

See Weaknesses.

---

> ### Author Response · Authors · 2023-11-23
> **Response Part 1**
>
> > The authors claim that Transformers demonstrate ...
>
> Thank you for your appreciative remarks and constructive feedback. It is noteworthy to highlight that our transformer architecture diverges from the conventional method of calculating attention maps, which involves attention across all pairs of nodes, resulting in a computational complexity of O(N^2). Instead, our approach involves attention exclusively across embeddings of the same nodes generated in different patches. Consequently, the total pairs of nodes for which attention is calculated is at most P squared, where P denotes the number of patches. This innovative approach is achieved through the unique combination of patchifier, local dynamic graph encoder, and Transformer, facilitating the pooling of embeddings for the same node.
> A more comprehensive analysis of computational complexity has been appended to Appendix A.4.1, providing a detailed calculation of the computational complexity of Todyformer based on the number of patches, edges, and nodes.
> It is also worth mentioning we have not stated the todyformer has computational capacity over  Graph Neural Networks (GNNs). In the Appendix A.4.2, we conducted a quantitative comparison of inference time versus performance for our model against baseline models. The results reveal that the new inference time is slightly higher than TGN but significantly lower than CAW, while our model's performance consistently outperforms all the compared models.
>
> > comment 2
>
> Our approach shares commonalities with Vision Transformer (ViT) up to the stage where both architectures follow a sequence of patchifying, tokenization, and Transformer utilization. However, distinctions arise in the patchifying process; we generate patches across the temporal domain, whereas ViT does so across the spatial domain. Moreover, tokenization in our model is achieved through local message-passing, in contrast to ViT's method of linear projection applied to flattened patches.
>
> We have conducted a comprehensive analysis and established certain levels of expressiveness through the lens of Weisfeiler-Lehman (WL) tests. The detailed proof and additional findings will be presented in the extended version of this work and in our forthcoming research. Our intention is to draw comparisons with the PINT model \cite{}, which has been theoretically established as expressive. This comparative analysis aims to contribute to a deeper understanding of the expressive capabilities of our model in relation to existing theoretical frameworks.
>
> > comment 3
>
> They have recently revised their pre-print from version 1 to version 2, as indicated by the ArXiv versioning system. In Table 2 and Table 10 , we have accordingly adjusted the main baseline values by incorporating the newly reported values from their revised paper. Notably, even with the updated baseline values, the performance of Todyformer remains superior to that of the other baselines.
>
> > comment 4
>
> The observed improvement in validation performance for two small datasets following the table update underscores the sensitivity of the final performance to random seeds. This reiterates the importance of comparing average performance across multiple trials to provide a more robust and representative assessment of the model's capabilities. Evaluating the mean performance helps mitigate the impact of random variations and provides a more reliable estimate of the model's overall effectiveness.  For TGBWiki and TGBReview, Todyformer achieves the second and third-best positions, respectively, on the test set. The observed performance suggests the possibility of overfitting on these smaller datasets. To address this, further exploration through hyperparameter search is recommended. Conducting an extensive search for optimal hyperparameters can help mitigate overfitting issues and enhance the model's generalization across diverse datasets.
>
> > comment 5
>
> We have updated our results by incorporating the standard deviation of the multi-trial outcomes into both Table 2 and Table 10. These values are based on 5 trials, utilizing the same negative samples provided by the leaderboard. The updated results have been seamlessly integrated into the paper and are thoroughly discussed to provide a more comprehensive understanding of the experimental outcomes. The protocols employed in the leaderboard adhere to the utilization of identical seeds and negative samples across methods. In the ranking process, the mean and standard deviation are pivotal metrics. Notably, our method demonstrates a significant distinction, particularly showcasing the lowest standard deviation for the three big datasets of TGBL. This highlights the robustness and consistency of our approach across multiple trials and reinforces its favorable performance characteristics.

---

### Official Review · Reviewer_Nqpw · 2023-10-30

**Soundness:** 3 good
**Presentation:** 2 fair
**Contribution:** 2 fair
**Rating:** 3
**Confidence:** 4

**Summary:**

This paper proposed a novel graph transformer method for dynamic graph. This model is an encoder-decoder architecture, started from patch generation and based on alternating between local and global message-passing as graph transformer. Authors perform two downstream tasks including future link prediction and dynamic node classification.

**Strengths:**

(1). This paper presents a graph transformer method on the dynamic graph, with enough experiments and ablation to show its performance.
(2). The purpose of using graph transformer is clearly discussed in Section Introduction and Related Work.

**Weaknesses:**

(1). Some presentations need improvement. Some notations need to be clear and some formulas need to be written for clarity. For example, what is the notation a,b means in E = |{e_a,...,e_b}| in section 3.2? What is the specific formula for positional encoding o() in section 3.4? For the transformer formula (6), could you specify whether there are LayerNorm and Feed-Forward modules as transformer? Could you give the formula of Q, K, V, and their dimension for clarity?

(2) I wonder about the results of Wikipedia and Reddit you mentioned in Section 4.1 datasets as it’s not shown in Table 1. As for the results shown in the Appendix, it seems they are not strong enough in the Inductive setting, especially for the Reddit dataset, which makes the statement in section 4.2“Todyformer consistently outperforms the baselines’ results on all datasets.” misleading.

**Questions:**

(1). In Section 3.2 PATCH GENERATION, could you please give more analysis of why you use this patch generation method, rather than other existing methods such as METIS in Graph ViT/MLP-Mixer or Graph coarsening in COARFORMER? What’s the advantage of your patch generation method?

(2). Section 3.5 “This issue is magnified in dynamic graphs when temporal long-range dependencies intersect with the structural patterns.” Could you please give some analysis of it or give some examples or literature to show the importance of over-smoothing problems in dynamic graphs? This can make this paragraph more convincing.

(3). Could you please show your efficiency comparison against other methods, especially CAW and Dyg2Vec? In my opinion, the computation of graph transformer on each node could have high time complexity, could you please analyze it?

(4) Could you give more analysis for the experiments weaker than the baseline? For example, in the second set of experiments, why did this method fail in these two smaller datasets?

---

> ### Author Response · Authors · 2023-11-23
> **Response Part 1**
>
> >  Some presentations need improvement. ..
>
> Acknowledged in full. We have revised the formulations within the main body of the text, denoting the modifications with the color blue for clarity and easy tracking. Additionally, the notation pertaining to edges in section 3.2 has been rectified. To enhance transparency, the positional encoding formulation has been added to the appendix and referenced in the methodology section (3.4). Detailed configurations of the transformer are now shown in the paper, revealing the presence of two transformer layers within each block, incorporating layer normalization and feed-forward modules. Notably, the dropout rate and number of heads for the transformer component are specified as 0.1 and 2 respectively. The dimensions of Q, K, and V for the transformer have been incorporated, accompanied by an explanation of how the remainder of the architecture is fed into the transformer component. To enhance clarity, we have included the formulas for Q, K, and V, along with their corresponding dimensions.
>
> > I wonder about the results of Wikipedia and Reddit you mentioned in Section 4.1...
>
> we appreciate your acknowledgment of the nuanced aspect requiring additional clarification in the paper. In the experimental setup, we have explicitly differentiated the names of the datasets employed in the first round of experiments from Reddit and Wikipedia. Furthermore, we have addressed and rectified potentially misleading terms in section 4.2, providing clarification and utilizing the color blue for emphasis and easy identification. Moreover, the discourse on the model's performance across the extended set of datasets has been incorporated into Appendix A.3.1.
>
> > In Section 3.2 PATCH GENERATION, ...
>
> In our study, we focus on temporal graphs, where the temporal domain adheres to Euclidean principles, allowing us to systematically generate patches, akin to other domains such as vision. Our choice of patchification strategy is motivated by its cost-effectiveness, avoiding unnecessary overhead and computational expenses. It's important to note that conventional approaches like METIS, which you referenced, employ structure-wise partitioning algorithms. However, these methods overlook the evolutionary nature of the temporal graphs we address in our work, rendering them suitable only for static graphs. Exploring the incorporation of both structural and temporal dimensions in our temporal patchification process is a promising avenue for future research, as it could enhance the overall understanding of the evolving graph dynamics.
>
> > Section 3.5 “This issue is magnified in dynamic graphs ...
>
> The problem definition for over-smoothing have been added in the paper. we have also added a section (A.5) to provide the results of experiments for over-smoothing and over-squashing.
>
> > Could you please show your efficiency comparison against other methods, especially CAW and Dyg2Vec? ...
>
> The qualitative and quantitative comparison of time complexity has been appended to Appendix A4. In A.4.1, an in-depth exploration of the model's time complexity concerning the number of nodes, edges, and patches is presented. In A.4.2, an analysis of Figure 4 is provided, depicting the performance versus inference time across three sizable datasets. Considering the delicate trade-off between performance and complexity, our models surpass all others in terms of Average Precision (AP) while concurrently positioning in the left segment of the diagrams, denoting the lowest inference time. Notably, as depicted in Figure 4, Todyformer remains lighter and less complex than state-of-the-art (SOTA) models like CAW across all datasets.
>
> > Could you give more analysis for the experiments weaker than the baseline? ...
>
> We have updated our results by incorporating the standard deviation of the multi-trial outcomes into both Table 2 and Table 10. These values are based on 5 trials, utilizing the same negative samples provided by the leaderboard. The observed improvement in validation performance for two small datasets following the table update underscores the sensitivity of the final performance to random seeds. This reiterates the importance of comparing average performance across multiple trials to provide a more robust and representative assessment of the model's capabilities. Evaluating the mean performance helps mitigate the impact of random variations and provides a more reliable estimate of the model's overall effectiveness.  For TGBWiki and TGBReview, Todyformer achieves the second and third-best positions, respectively, on the test set. The observed performance suggests the possibility of overfitting on these smaller datasets. To address this, further exploration through hyperparameter search is recommended. Conducting an extensive search for optimal hyperparameters can help mitigate overfitting issues and enhance the model's generalization across diverse datasets.

---

### Official Review · Reviewer_1VfV · 2023-10-30

**Soundness:** 3 good
**Presentation:** 2 fair
**Contribution:** 3 good
**Rating:** 6
**Confidence:** 3

**Summary:**

This paper proposes TODYFORMER, a novel Transformer-based neural network tailored for dynamic graphs. It unifies the local encoding capacity of Message-Passing Neural Network with the global encoding of Transformers.
Experimental evaluations on public benchmark datasets demonstrate that Todyformer consistently outperforms the state-of-the-art methods for the downstream tasks.

**Strengths:**

1. The paper is clearly written and easy to follow.
2. The results of different downstream task prove the validity and superiority of model, especially the results on large scale datasets.

**Weaknesses:**

1.	This paper does not provide the mathematical form of the overall loss function, which leads to an incomplete explanation of the model in Section 3.
2.	In the detail of three main components, many ideas are not novel. For instance, in encoding Module, the Transformers is very basic model. The window-based encoding paradigm is from DyG2Vec. The positional-encoding idea is also from previous work.

**Questions:**

1.	Could you add mathematical form of the overall loss function?

---

> ### Author Response · Authors · 2023-11-23
> **Response Part 1**
>
> Thank you for providing  the positive comments and constructive feedback.
> > 1. This paper does not provide the mathematical form of the overall loss function, which leads to an incomplete explanation of the model in Section 3.
>
> To elucidate the precise loss functions employed in our study for various downstream tasks, we have incorporated the mathematical formulations of these losses into the appendix of the main manuscript. Furthermore, to provide a comprehensive understanding of our approach, we have included references to prior works that have utilized similar loss functions. This ensures transparency in our methodology and facilitates a contextual understanding of the chosen loss functions in the broader scientific discourse.
>
> >  In the detail of three main components, many ideas are not novel. For instance, in encoding Module, the Transformers is very basic model. The window-based encoding paradigm is from DyG2Vec. The positional-encoding idea is also from previous work.
>
> In response to the concern regarding the perceived lack of novelty in our work, we have explicitly emphasized the innovative aspects of our research both within the revised manuscript and in this rejoinder. While it is acknowledged that the individual components we employ share nomenclature with existing literature in fields such as Computer Vision and Natural Language Processing, our study marks the inaugural amalgamation of these components. This unique integration is specifically tailored to enhance the expressive capabilities of dynamic graph encoders. Each constituent has been purposefully incorporated to address the inherent limitations of dynamic graph encoders, thus contributing novel insights to the field.
>
> >  Could you add mathematical form of the overall loss function?
>
> Noted, the mathematical formulations have been added to the appendix and it's referenced in the main text.

---

### Meta-Review · Area_Chair_kwcZ · 2023-12-09

**Metareview:**

In this paper, the authors propose Todyformer, a tokenized graph Transformer for dynamic graphs, where over-smoothing and over-squashing are empirically improved through a local and global encoding architecture. The primary novel components are patch generation, structure-aware tokenization using typical MPNNs that locally encode neighborhoods, and the utilization of Transformers to aggregate global context in an alternating fashion. The experimental results show the effectiveness of the proposed method.

The paper is well-written. However, there are many concerns from the reviewers about the novel and the motivation of the proposed approach, missing some key comparisons and experimental reports, as well as technical details of the paper. The authors addressed them during the rebuttal, but many concerns are still not solved completely.

**Justification For Why Not Higher Score:**

Many concerns about the paper are still not solved completely

**Justification For Why Not Lower Score:**

N/A

---

### Decision · Program_Chairs · 2024-01-16

Reject